# Patterns of microbial communities were shaped by bioavailable P along the elevation gradient of Shergyla Mountain, as determined by analysis of phospholipid fatty acids

Duo Ba[1], Duoji Qimei[2], Wei Zhao[3], Yang Wang📷[4]*

1 Bureau of Ecology and Environment of Naqu City, Tibet Autonomous Region, Lhasa, China, 2 Tibet University, Lhasa, China, 3 Department of Ecology and Environment of Tibet Autonomous Region, Lhasa, China, 4 Xizang Autonomous Region Development and Reform Commission, Lhasa, Tibet Autonomous Region, China

* yangwang1970@outlook.com

**Data Availability Statement:** All relevant data are within the paper and its Supporting Information files.

## Abstract

The distribution pattern of the microbial community in mountains is an important component of biodiversity research. Many environmental factors vary significantly with elevation on a relatively small scale in subalpine and alpine environments. These factors may markedly affect microbial community composition and function. In this study, we analyzed phospholipid fatty acid (PLFA) profiles and phosphorus (P) fractions in soils from 9 sites along an elevation gradient (3500–4100 m above sea level (a.s.l.)) of the Shergyla Mountain, Tibet in China. Many biomarker PLFAs indicated that there were biogeochemical trends of the microbial distribution patterns of some soil microorganisms, which were most often increasing, U-shaped and unimodal trends along the elevation gradient. A redundancy analysis (RDA) and correlations indicated that P factors (e.g., Resin-Pi, NaHCO$_3$-Pi and NaHCO$_3$-Po) were more important in controlling the microbial PLFA distribution pattern than other factors (e.g., MAT, MAP, pH, TOC, TN and soil moisture) in this study area. Microorganisms are strongly associated with P fractions. Our results suggested that microbial communities were subjected to P stresses and that the distribution patterns of microbial communities were shaped by bioavailable P along the elevation gradient. Our work also hints that P geochemical processes drive the microbial diversity of the Shergyla Mountains.

## Introduction

Microbial diversity and community composition are important parameters in the study of soil microorganisms. Despite the large number of soil microbial species, it was thought that there is no microbial biogeography because of the high propagation and dispersal rates of microbes. However, some researchers call the conception into question and begin to investigate microbial biogeography at different scales. At small spatial scales in soil, a significant pore-scale microbial biogeography was observed [1]. At large spatial scales, the soil microbial community

**Funding:** This work was funded by intergovernmental cooperation in science, technology and innovation, and by the National Key Research and Development Program (2019YFE0194000), the National Natural Science Foundation of China (42077013).

**Competing interests:** The authors have declared that no competing interests exist.

structure displayed some biogeographical distribution trends with latitude in eastern China [2]. Research on the bacterial diversity of soil samples collected from across North and South America has suggested that microbial biogeography is controlled primarily by edaphic variables [3]. Microbial biogeography has rarely been mentioned or studied in alpine and subalpine areas. In particular, alpine regions are ideal places to inform our understanding of how microbial biogeography is affected by environmental factors. There is a large elevation change in mountain ecosystems, which results in various environmental gradients, including a climatic gradient, vegetation belts and soil succession. These natural successions along latitude gradients are often spread over thousands of kilometers, but they may occur along elevation gradients within tens of kilometers. The interactions between vegetation and soil may impact the adaptation of microbial communities to mountain environments [4]. It is necessary to know how soil microorganisms adapt to the environment along elevation gradients because this knowledge will contribute to our understanding of microbial biogeography, especially in the context of global climate change. Additionally, findings about the effects of elevation on microbial biogeography may be applied to the related study of the effects of latitude.

Phosphorus (P) and other nutrients are essential for microbial growth [5]. In contrast to carbon and nitrogen, which have atmospheric sources, P in a terrestrial ecosystem ultimately originates from the slow weathering of soil minerals. Available P is reactive and easily fixed by soil components, including Fe, Al and Ca hydroxides and other compounds. Therefore, P becomes a limiting factor in some terrestrial ecosystems. Soil microbial processes are believed to be especially limited by P in the majority of tropical forests in older soils [6, 7]. Kunito *et al.* [8] reported that microbial communities might be P-limited because they found high phosphodiesterase and phosphomonoesterase activities in Japanese forest soils. Soil microbial biomass was also affected by P in some warm-temperate forest soils [9]. In addition, in temperate forest soils in the Dysart Woods, Deforest and Scott [10] demonstrated that the bicarbonate organic P (Po) fraction had the strongest influence on microbial community composition, which was similar to the influence of soil pH. However, Groffman and Fisk [11] demonstrated that soil microorganisms were not sensitive to the addition of phosphate in a northern hardwood forest.

Recently, the microbial community composition and the factors that influence it were investigated in undisturbed mountain areas by several researchers. Margesin *et al.* [12] investigated microbial communities and activities in alpine and subalpine soils in the central Austrian Alps. Djukic *et al.* [13] compared soil nutrition and microbial community compositions in different alpine vegetation zones. Männistö *et al.* [14] focused on the pattern of bacterial community compositions at different altitudes. These investigations and other studies have shown that soil microbial community composition, activity and biogeography are strongly affected by soil pH [3, 15, 16]. Additionally, there is a significant relationship between P and pH in soils [17]. It is well known that soil pH influences P solubility and uptake [18]. Generally, the secretions of soil microorganisms and roots can change the soil pH to obtain enough P nutrients to support their growth [19, 20]. Therefore, we inferred that there are important correlations between patterns of microbial community structure and soil P along the elevation gradient. Thus, some important questions to investigate are as follows: How is the microbial community structure influenced by P in alpine and subalpine terrestrial ecosystems?

The traditional method for studying soil microbial communities has utilized culture-dependent techniques that rely on the isolation of microbes from soil samples (e.g., microbial counts and membrane-filtration techniques). This method provides limited information on the microbial community because the majority of soil microorganisms cannot be cultured under laboratory conditions [21]. In addition, enzymatic and metabolic activities in the soil have been used to estimate the pattern of soil microbial community compositions (e.g., Biolog

microtiter plates) [22]. Recently, molecular methods that rely on molecules such as DNA, RNA and PLFAs have been widely used to characterize soil microbial communities in the natural environment. PLFAs are the major components of microbial cell membranes and decompose rapidly after microorganisms die. Additionally, different PLFAs can distinguish different groups of species of soil microorganisms and are thus commonly considered to be a "signature" for specific microbial groups [23]. The analysis of PLFAs has produced a significant amount of valuable information, including the distribution, functional components and influencing factors of microbial communities [24, 25]. It has been demonstrated that PLFA analysis is a suitable and powerful method for analyzing microbial community composition [12].

The objective of this study was to determine the distribution pattern of soil microorganisms and the relationship between P and microbial community structure at subalpine and alpine sites in Shergyla Mountain, Tibet, in China. Methods including microbial PLFA identification and sequential P extraction were used to elucidate the relationship between PLFAs and P. We tested the hypotheses that (1) biogeography exists in soil microbial communities along altitudinal belts of mountain areas and that (2) microbial community composition is significantly correlated with P speciation in mountain areas. This study also attempted to provide information on the microbial communities present in undisturbed soils, which may contribute to an understanding of the data of other studies on disturbed soils, such as managed or degraded soils.

## Materials and methods

### Ethics statement

All necessary permits were obtained for the described field studies. We conducted this study at Shergyla Mountain, which is under the jurisdiction of the People's Government of Nyingchi City, Tibet Autonomous Region, People's Republic of China. We obtained permission from the People's Government of Nyingchi City to use the sample plots. Furthermore, our study did not harm the environment and did not involve endangered or protected species.

### Study area

The study area was located in Shergyla Mountain (93˚12' — 95˚35' E, 29˚10' — 30˚15' N), Tibet, China. Shergyla Mountain is located in the southeastern Tibetan Plateau with a summit elevation of 5300 m above sea level (a.s.l.), and. The climate at Shergyla Mountain is dominated by the Indian monsoon, with a monthly mean temperature between -14.0˚C in January and 9.2˚C in July. The mean annual precipitation (MAP) is 985 mm, and the mean annual temperature (MAT) is 3.4˚C at 3900 m a.s.l., while the MAP is 841 mm, and the MAT is 4.9˚C at 3326 m a.s.l.

Shergyla Mountain has an intact and continuous vertical vegetation spectrum. The primordial forest and meadow are distributed with elevation. From 2800 to 3500 m a.s.l., the broadleaved and dark coniferous mixed forest zone is dominated by *Quercus aquifolioides*, *Picea likiangensis var. Linzhiensis* and *Pinus densata*. The coniferous forest zone is distributed from 3500 to 4300 m a.s.l. and is dominated by *Abies georgei var. smithii* and *Picea likiangensis var. Linzhiensis*. The shrub zone is from 4300 to 4600 m a.s.l., where the dominant plants are *Rhododendron nivale*, *Cassiope fastigiata* and *Rhododendron lepidotum*. The alpine meadow zone spans from 4600 m a.s.l. to the snowline. The present study zone was from 3500 to 4100 m a.s.l. Details of the zones and soil properties are presented in Table 1.

### Soil sampling

In July 2020, soil samples were individually collected from the subalpine and alpine zones, including nine elevation gradients (from 3500 to 4100 m a.s.l) on Shergyla Mountain

**Table 1. Soil properties and climatic factors in this study area.**

| Elevation (m a.s.l) | MAT (°C) | MAP (mm) | TOC (g/kg) | TN (g/kg) | TP (mg/kg) | pH | Soil moisture (%) | Vegetation zone |
|---|---|---|---|---|---|---|---|---|
| 4100 | 2.2 | 990.5 | 43.8±2.6a | 2.64±0.29a | 711.1±39.9ab | 4.98±0.03a | 51.5±6.0a | Coniferous forest zone |
| 4060 | 2.4 | 989.3 | 47.9±1.5a | 3.34±0.45a | 533.5±5.9b | 5.13±0.15a | 47.0±1.5a | Coniferous forest zone |
| 3980 | 2.9 | 987.0 | 44.1±2.3a | 3.63±0.2a | 390.9±5.4a | 4.91±0.12a | 45.9±1.4a | Coniferous forest zone |
| 3930 | 3.3 | 985.5 | 48.4±4.0a | 3.14±0.09a | 393.9±25.5ab | 5.04±0.10a | 48.7±2.4a | Coniferous forest zone |
| 3910 | 3.4 | 984.9 | 52.4±5.4a | 3.1±0.25a | 377.7±8.5a | 5.06±0.25a | 54.6±1.2a | Coniferous forest zone |
| 3780 | 3.9 | 944.6 | 52.9±6.3a | 3.42±0.30a | 436.1±47.8ab | 4.90±0.16a | 56.8±2.0a | Coniferous forest zone |
| 3700 | 4.2 | 919.5 | 47.0±1.6a | 3.22±0.34a | 412.2±29.0ab | 5.12±0.15a | 51.2±5.9a | Coniferous forest zone |
| 3590 | 4.7 | 885.0 | 40.3±2.6a | 3.17±0.14a | 501.3±47.6ab | 5.25±0.09a | 39.7±8.3a | Coniferous forest zone |
| 3500 | 5.1 | 856.8 | 54.9±2.9a | 3.95±0.11a | 808.5±66.5ab | 4.79±0.11a | 54.5±6.9a | Coniferous forest zone |

MAT: mean annual temperature; MAP: mean annual precipitation. Dominant plants in coniferous forest zone are *Abies georgei var.smithii* and *Picea likiangensis var*. *Linzhiensis*. Different letters indicate significant differences ($p < 0.05$)

(Table 1). At each elevation, three plots (20 m × 20 m, >30 m intervals between the plots) were set up according to the "Protocols for Standard Soil Observation and Measurement in Terrestrial Ecosystems" [26]. Along one diagonal line of each plot, we randomly chose six sampling points (more than 2 m intervals between sampling points). At each sampling point, the litter layer and roots were removed before the soil samples were collected using a sterile blade at a depth of 0–10 cm. To obtain a representative soil sample for each plot, soil samples from six sampling points of the plot were homogenized based on the same weight and were stored in coolers containing bags of ice. Finally, twenty-seven soil samples were obtained for the nine elevations. When the soil samples were taken back to the laboratory, large dopants (e.g., large stones or litter) were removed, the soil samples were screened with 2 mm sieves, and soil moisture was measured immediately. A subsample was freeze-dried and stored at -18°C until PLFA analysis. Another subsample was air-dried for physical and chemical analyses.

## Phospholipid fatty acid analysis

PLFAs were extracted according to the procedure described by Wu *et al.* [27]. Briefly, freeze-dried soil samples (3 g) were extracted using a single-phase mixture of chloroform:methanol: citrate buffer (30.4 ml at a 1:2:0.8 volume ratio). After phase separation, the $CHCl_3$ layer (extracted lipids) was collected and dried under $N_2$ at 30°C. The dry residue was transferred into a silica solid phase extraction column (3 ml standard SPE tube, Supelco Inc., Bellefonte, PA), and neutral lipids and glycolipids were removed by sequential elution with chloroform (10 ml) and acetone (10 ml). Phospholipids were then collected by elution with methanol (10 ml) and dried under $N_2$. Afterward, the phospholipid fraction was methylated with a methanol-toluene solution and a potassium hydroxide methanol solution, and $H_2O$ (2 ml) and acetic acid (0.3 ml) were added. Fatty acid methyl esters were extracted in hexane (2×2 ml) and dried under $N_2$. Samples were analyzed on an Agilent 6890 N Gas Chromatograph using MIDI peak identification software (Version 4.5; MIDI Inc., Newark, DE). The column was an Agilent 19091B-102 (25.0 m×200 μm×0.33 μm) capillary column, and $H_2$ was used as the carrier gas. The GC temperature program was set by MIDI software. Fatty acid 19:0 was used as an internal standard and added to samples before methylation. Identification and quantification of fatty acid methyl esters were conducted automatically by MIDI peak identification software.

Fatty acids are designated according to the nomenclature described by Petersen and Klug [28]. The total PLFA concentration was calculated using all of the PLFAs detected (44 PLFAs). The fatty acids 14:0, 15:0, i14:0, a15:0, i15:0, i16:0, a17:0, i17:0, 10me16:0, 10me17:0, 16:1ω7c,

cy17:0 and cy19:0 were used as bacterial biomarkers [25, 29, 30]. PLFAs 18:1ω9c and 18:2ω6,9c were chosen to represent fungi [31, 32]. A marker for actinomycetes is 10me18:0 [33]. The PLFA 16:1ω5c was used as a marker for arbuscular mycorrhizal fungi (AMF) [34]. The PLFAs i14:0, i15:0, i16:0, i17:0, a15:0 and a17:0 were chosen to represent Gram-positive bacteria [32, 35]. The PLFAs 14:0, 15:0, 16:1ω7c, cy17:0 and cy19:0 were chosen to represent Gram-negative bacteria [13, 36]. The cyclopropyl:precursor (cy:pre) ratio was calculated using (cy17:0 + cy19:0)/(16:1ω7 + 18:1ω7). The PFLA ratio (i.e., cy:pre) was used as an indicator of environmental stress [2, 37, 38].

## Quantification of P fractions

Sequential P chemical extraction of the soil was conducted according to Hedley *et al.* [39] and Agbenin *et al.* [40]. Briefly, 0.5 g of oven-dried soil was weighed and placed into a centrifuge tube. After the tube was shaken for 16 h with 30 ml of deionized water and two resin strips (area: 1 cm × 3.5 cm, Anion 204UZRA), the Resin-Pi was extracted and measured. Then, 0.5 M $NaHCO_3$ was added to the tube to extract the inorganic and organic P, $NaHCO_3$-Pi and $NaHCO_3$-Po, respectively. $NaHCO_3$-Pi, $NaHCO_3$-Po and Resin-Pi are considered the most bioavailable P fractions [41]. To extract the inorganic and organic phosphorous fractions (NaOH-Pi and NaOH-Po, respectively), which were less labile and adsorbed onto Fe or Al oxides and humic compounds in the soils, 0.1 M NaOH was added to the tube [42]. Next, 1 M HCl was added to the soil remaining from the previous step to extract HCl-Pi and HCl-Po. After extraction with 1 M HCl, concentrated hydrochloric acid (conc. HCl) was added to the remaining soil to extract conc. HCl-Pi and conc. HCl-Po, which included apatite and more stubborn phosphates. The total P (TP) present in the soil was determined by summing all of the P fractions. All extracts were centrifuged at 8000 rpm for 10 min. For all of the above extracts, the inorganic P ($PO_4^{3-}$-P) present in the extract was determined directly using the Murphy and Riley method [43], and the TP (including organic and inorganic P) was determined after persulfate digestion [44] using the Murphy and Riley method. The organic P content in each extract was determined by calculating the difference between the total P and inorganic P contents.

## Soil physical and chemical analyses

All analyses were conducted on the basis of the technical regulations of the Soil Science Society of China [45]. Briefly, soil total organic carbon (TOC) was determined using dichromate digestion. Soil moisture was measured gravimetrically by oven-drying the soil to a constant weight at 105˚C. Soil pH (soil:water = 1:2.5) was determined using a pH meter. Soil total nitrogen (TN) was determined using the Kjeldahl determination method. According to Tabatabai [46], the activities of acid phosphomonoesterase were assayed with *p*-nitrophenyl phosphate (PNPP, Sigma N4645) as substrates at pH 6.5. The mean annual temperature (MAT) and mean annual precipitation (MAP) with elevation were obtained by fitting the climatic data from Luo et al. [47].

## Statistical analysis

All analyses were based on the relative abundance (%) of individual fatty acids except for the total PLFA content (nmol $g^{-1}$, on a dry weight basis). Statistical analyses were conducted using SPSS 13.0 software for Windows. Spearman's rank correlation method was used to determine correlations between the measured properties. All sites were compared with a one-way analysis of variance (ANOVA). If the variance of the variable was homogeneous, Tukey's post-hoc tests

were used for multiple comparisons within the ANOVA; otherwise, the Games-Howell tests were used.

A quadratic polynomial model (or a linear regression model) was used to model the distribution pattern of biomarker PLFAs with elevation (Fig 1). The redundancy analysis (RDA) was calculated and graphed using CANOCO for Windows (version 4.5). This analysis was used to visualize the relationships between the environmental variable gradients, the response variable values and the samples [38]. In this study, P fractions, soil property data and climate data were used as the environmental variables, and the PLFA data were used as the response variables (referred to as 'species' by the CANOCO program) after the PLFA data were transformed using a log ratio (Y' = log(Y+1)). In the CANOCO program, the interspecies correlations were chosen as the focus of the ordination in the analysis, and species scores were divided by the standard deviation to reduce the effects of extreme values. The ordination was centered using species and samples. The significance of the relations between the ordination and explanatory variables was tested using a permutation test with 499 permutations. Moreover, we used partial RDA to tease apart the pure effects of P fractions and other factors (i.e., MAT, MAP, TOC, TN, soil pH and soil moisture) on the PLFA matrices. For instance, the P fractions or other factors matrix was the explanatory matrix, and the other one was the partial matrix. Partial RDA examined, for example, the influence of P fractions on the PLFA abundances while controlling for other factors, and vice versa. More details on the RDA can be found in Sun *et al*. [48] and terBraak [49].

Nonmetric multidimensional scaling (NMDS) analysis based on Bayesian distances was performed to analyze the overall pattern of microbial PLFA distribution across alpine and subalpine zones (low-elevation: 3500–3700 m a.s.l., mid-elevation: 3780–3930 m a.s.l., and high-elevation: 3980–4100 m a.s.l.). The significance of the pattern was examined using an analysis of similarities (ANOSIM) with $p < 0.05$. They were performed using the "vegan" package of the R software environment [50].

## Results

### Basic soil properties

All of the studied soils were generally acidic, with similar pH values along the elevation gradient (Table 1). Changes in the pH were small, with a range of 4.79–5.25 and a mean of 5.02. No significant difference was found in pH between elevations ($p > 0.05$). The TOC, TN and soil moisture did not display a significant trend along the elevation gradient ($p > 0.05$). The TOC mainly ranged from 40.3 g/kg to 54.9 g/kg, and the TN in the soils varied from 2.64 g/kg to 3.95 g/kg. The soil moisture values were high, with a range of 39.7–56.8% and a mean value of 50.0% for all elevations. In contrast, soil TP had greater variability along the elevation gradient and showed a significant U-shaped distribution trend ($p < 0.05$).

### The P fractions at the studied sites

Our results showed that both Resin-Pi and NaHCO$_3$-Pi were at low levels in the topsoil of the coniferous forest zone (Table 2). The Resin-Pi concentrations ranged from 0.48 to 0.57 mg/kg, with an average value of 0.54 mg/kg. The NaHCO$_3$-Pi concentrations ranged from 0.41 to 0.49 mg/kg, with an average value of 0.46 mg/kg. Resin-Pi, NaHCO$_3$-Pi and NaHCO$_3$-Po are considered bioavailable P [41]. The bioavailable P concentrations (Resin-Pi + NaHCO$_3$-Pi + NaHCO$_3$-Po) ranged from 1.86 to 2.33 mg/kg. The bioavailable P constituted 0.18–0.56% of the TP for all sites. NaHCO$_3$-Po was the major component accounting for 53% (ranging from 0.99 to 1.25 mg/kg) of bioavailable P. Among the different elevations, the three bioavailable P showed significant differences. NaOH-Pi and NaOH-Po are

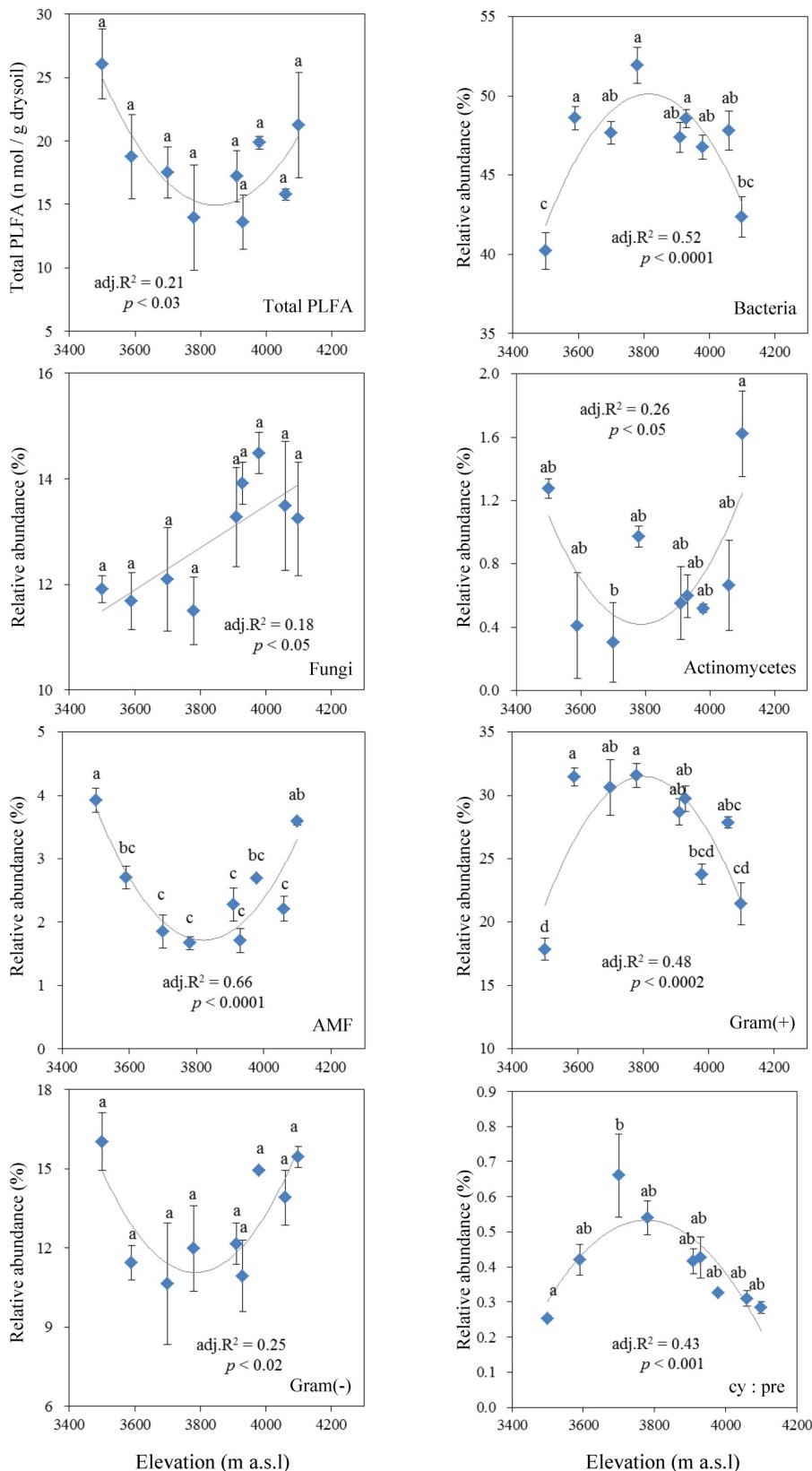

**Fig 1. Sums and ratios of PLFA from various microbial groups along the elevation gradient.** Values are arithmetic means ± standard errors (n = 3). Different letters in the bars indicate significant differences ($p < 0.05$). AMF: arbuscular mycorrhizal fungi, Gram(+): Gram-positive bacteria, Gram(-): Gram-negative bacteria, cy:pre ratio was calculated using (cy17:0 + cy19:0)/(16:1ω7 + 18:1ω7).

considered less bioavailable P because they are adsorbed onto Fe or Al oxides and humic compounds in soils [51]. Compared with the bioavailable P, NaOH-Pi and NaOH-Po exhibited great spatial variations within the coniferous forest zone. The lowest value of NaOH-Pi (0.67 mg/kg) was detected at 3780 m a.s.l., and the highest value (1.17 mg/kg) was detected at 3590 m a.s.l. (Table 2). In particular, NaOH-Po showed great variations (ranging from 0.08 to 25.05 mg/kg, with a mean of 10.01 mg/kg). Compared with other P fractions, HCl-Pi and CHCl-Po were at very high levels (343.7–748.8 mg/kg for HCl-Pi, 28.56–31.60 mg/kg for HCl-Po).

## Microbial community composition

Microbial PLFA composition, which is a measure of microbial community composition, showed that the microbial community compositions were obviously different and showed various distribution patterns along the elevation gradient, including increasing, U-shaped and unimodal trends (Fig 1). The total PLFA, the sum of the 44 fatty acids that were identified in the studied soils, showed a significant U-shaped distribution trend with elevation ($p < 0.03$ and adj. $R^2 = 0.21$), although the total PLFA concentrations were not significantly different between the different elevations ($p > 0.05$). Bacterial PLFAs comprised a large portion of the total PLFAs, ranging from 40% to 54%. The relative abundance of bacterial PLFAs had a significant unimodal pattern ($p < 0.0001$ and adj. $R^2 = 0.52$) along the elevation gradient, and the peak value was found at 3780 m a.l.s. Fungal PLFAs had a significant linear pattern ($p < 0.05$ and adj. $R^2 = 0.18$) along the elevation gradient but was not significantly different between different elevations. Actinomycete PLFA was higher at high-elevation sites and low-elevation sites than at mid-elevation sites, and a U-shaped distribution trend ($p < 0.05$ and adj. $R^2 = 0.26$) was found along the elevation gradient.

The PLFA 16:1ω5c, which is representative of AMF, accounted for a small portion of the total PLFAs and ranges from 1% to 4% at these sites (Fig 1). The relative abundance of AMF PLFAs also had a significant U-shaped distribution trend ($p < 0.0001$ and adj. $R^2 = 0.66$) along the elevation gradient and had its lowest value at 3780 m a.s.l. As the main component of the

**Table 2. Concentrations of P fractions with elevation.**

| Elevation (m a.s. l) | Resin-Pi (mg/kg) | NaHCO₃-Pi (mg/kg) | NaHCO₃-Po (mg/kg) | NaOH-Pi (mg/kg) | NaOH-Po (mg/kg) | HCl-Pi (mg/kg) | CHCl-Po (mg/kg) |
|---|---|---|---|---|---|---|---|
| 4100 | 0.520±0.004ab | 0.44±0.02 ab | 1.08±0.018ab | 1.10±0.05a | 16.95±10.67a | 661.1±48ab | 29.92±1.94a |
| 4060 | 0.567±0.005a | 0.47±0.01 ab | 1.17±0.03a | 1.08±0.088ab | 19.02±6.58a | 481.7±6.9a | 29.45±1.5a |
| 3980 | 0.568±0.007a | 0.49±0.01a | 1.19±0.02a | 0.87±0.01ab | 3.81±2.05a | 354.7±4.3b | 29.24±1.93a |
| 3930 | 0.559±0.009a | 0.48±0.01 ab | 1.19±0.02a | 1.09±0.09a | 0.89±0.65a | 361.1±25.3ab | 28.56±0.65a |
| 3910 | 0.556±0.012a | 0.47±0.02 ab | 1.16±0.04ab | 1.08±0.04ab | 0.08±0.02a | 343.7±8.2b | 30.63±0.48a |
| 3780 | 0.567±0.007a | 0.48±0.01a | 1.17±0.01a | 0.67±0.07b | 6.62±2.71a | 394.9±45.8ab | 31.59±1.53a |
| 3700 | 0.536±0.009a | 0.46±0.02 ab | 1.13±0.04ab | 0.86±0.09ab | 17.97±7.59a | 361.2±21.8ab | 30.03±2.17a |
| 3590 | 0.545±0.015a | 0.46±0.01ab | 1.12±0.026ab | 1.17±0.08a | 0.11±0.05a | 468.2±49ab | 29.72±1.52a |
| 3500 | 0.475±0.012b | 0.411±0.004b | 1.00±0.01b | 1.08±0.06ab | 25.05±5.11a | 748.8±64.1ab | 31.60±1.19a |

Values are means ± SE (n = 3), HCl-Pi = HCl-Pi + conc. HCl-Pi, HCl-Po = HCl-Po + conc. HCl-Po.

bacterial population, similar to bacterial PLFAs, the PLFAs ascribed to Gram-positive bacteria also displayed a significant unimodal pattern ($p < 0.0002$ and adj. $R^2 = 0.48$). For gram-positive bacteria, their PLFAs, by contrast, showed a significant U-shaped distribution trend with elevation ($p < 0.02$ and adj. $R^2 = 0.25$). The cy:pre ratio also has a significantly unimodal trend ($p < 0.001$ and adj. $R^2 = 0.43$).

## Relationship between microbial PLFAs and environmental factors

RDA based on PLFA data and environmental factors was performed (Fig 2A) to present all PLFA distributions and identify the main factors regulating microbial community composition in the natural mountain environment. The RDA diagram showed that most PLFAs were distributed in the first, third and fourth quadrants (Fig 2A). The environmental factors significantly explained the variation in the microbial PLFA composition ($p < 0.001$, Fig 2A). The RDA also indicated that P factors (e.g., Resin-Pi, NaHCO$_3$-Pi and NaHCO$_3$-Po) were more important in controlling the microbial PLFA distribution pattern than other factors (e.g., MAT, MAP, pH, TOC, TN and soil moisture) in this study area. Additionally, a partial RDA showed that P factors and other factors could explain 56% and 34% of the total variance in microbial PLFA composition, respectively (Fig 2B). NMDS analysis and ANOSIM indicated that the microbial PLFA composition showed significant clustering with elevation (Fig 2C).

The proportion of individual PLFAs that were selected as biomarkers also changed with the concentrations of P fractions (Table 3). To further explore the relationships between biomarker-PLFAs and soil P, a correlation analysis was performed (Table 3). The relative abundances of bacterial and fungal PLFAs were significantly correlated with bioavailable P concentrations (i.e., Resin-Pi, NaHCO$_3$-Pi and NaHCO$_3$-Po) ($p < 0.05$). The relative abundances of actinomycete, gram-positive bacterial and gram-negative bacterial PLFAs all correlated strongly with NaOH-Po and HCl-Pi ($p < 0.05$). AMF correlated negatively with Resin-Pi ($p < 0.05$), NaHCO$_3$-Pi ($p < 0.1$) and NaHCO$_3$-Po ($p < 0.05$).

# Discussion

## Microbial community distribution determined by PLFA with increasing elevation

The spatial distribution patterns of soil microbial species are generally considered to be cosmopolitan and random because of the biological properties of microbes (e.g., low extinction and speciation rates) and our lack of understanding of their diversity due to technical and conceptual reasons [52]. However, the existence of microbial biogeography has been demonstrated and confirmed by an increasing number of studies from the micro to the global scale [1, 53, 54]. Microbial distribution patterns have been studied using taxonomy and trait-based approaches to microbial biodiversity and biogeography [55]. Trait variation with site properties is generally studied to understand how vegetation properties shift along geographical gradients and to thus predict habitat boundaries in changing environments [55]. As an analogous method, PLFA profiling of microbial communities could be feasible for determining the biogeographic patterns of soil microorganisms. PLFA biomarkers are usually used to identify microbial taxa, and PLFA profiles are affected by the metabolic state of microorganisms [56].

In the present study, the PLFA from soil samples along an altitudinal gradient can be regarded as one trait of soil microbial communities. Trend regression analysis showed that a spatial pattern of soil microorganisms was visible along the elevation gradient (Figs 1 and 2). In agreement with our first hypothesis, this trend suggested that trait-based biogeography exists in the altitudinal belts of mountain areas. Similar to our results, Sun et al. [57] used 16S rRNA

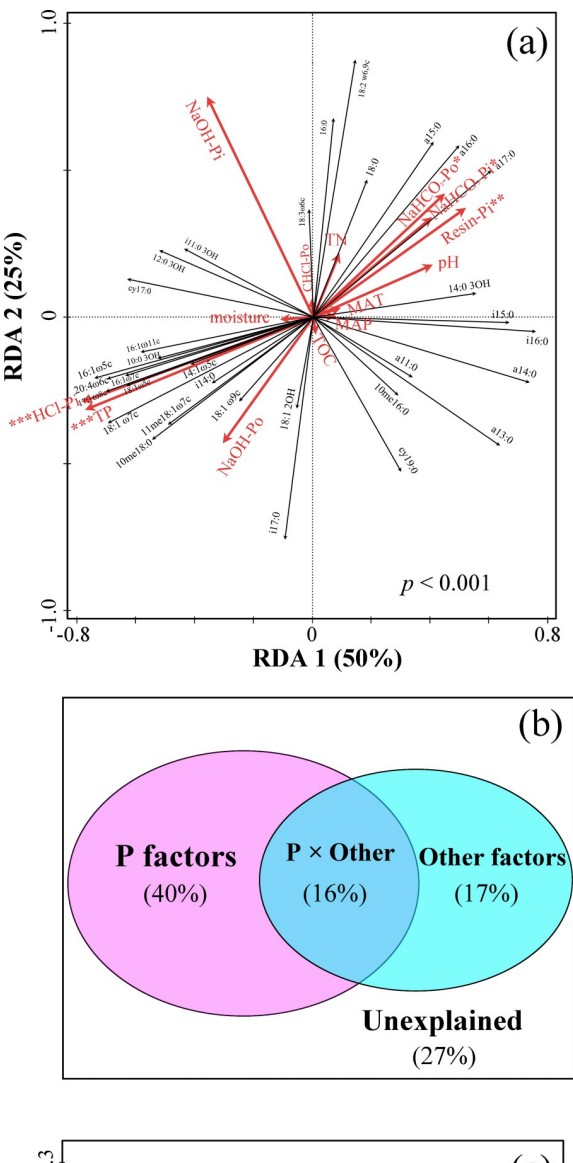

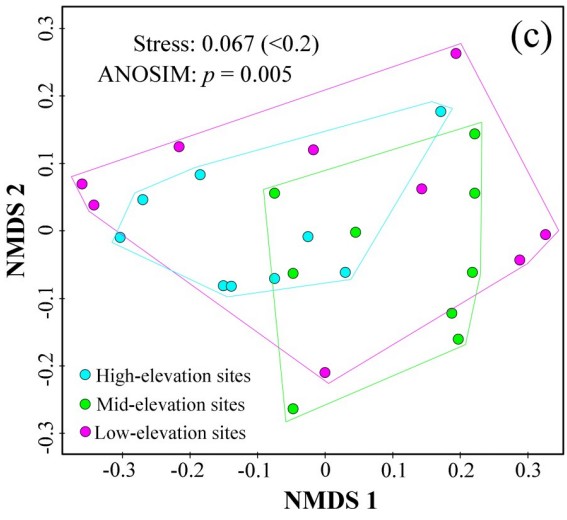

**Fig 2. Analysis of distribution patterns of soil microbial communities along the elevation gradient.** (a) Redundancy analysis (RDA) of the PLFA dataset for the 27 soil samples using 44 PLFAs as species and 14 environmental variables as explanatory variables. The significance of the relations between the ordination and explanatory variables is denoted as follows: $^*$ $p < 0.05$, $^{**}$ $p < 0.01$, $^{***}$ $p < 0.001$. (b) A partial RDA showed the proportion of the variance in PLFAs composition explained by the P factors (i.e., P fractions) and other factors (i.e., MAT, MAP, pH, TOC, TN and soil moisture). (c) Nonmetric multidimensional scaling (NMDS) analysis of microbial PLFAs across alpine and subalpine zones, ranked by elevation (low-elevation sites: 3500–3700 m a.s.l., mid-elevation sites: 3780–3930 m a.s.l., high-elevation sites: 3980–4100 m a.s.l.).

sequencing to reveal that target bacterial communities were distinctly different among the elevations, and most of these bacteria showed various regular distribution patterns with increasing elevation (from 3000 to 4300 m a.s.l.), including increasing, hump-backed and hollow trends. Their research showed the biogeographic pattern of soil microbial distribution with increasing elevation at Gongga Mountain. Although our study and theirs used different methods to measure microbial communities, both studies revealed similar patterns of microbial distribution along the elevation gradient. The similarity of the results demonstrated that the analysis function of inexpensive PLFA technology is similar to that of 16S rRNA sequencing for the biogeographic pattern of microbial distribution. Not coincidentally, at the continental scale, Fierer and Jackson [3] clearly indicated that microbial biogeography was controlled primarily by edaphic variables and differed fundamentally from the biogeography of "macro" organisms. In particular, they attributed the biogeographic pattern of microbial distribution mainly to soil pH. Similarly, a study from Changbai Mountain also attributed the elevational patterns of microbial community composition to soil pH [58]. In addition, some studies have suggested that vegetation type, soil organic matter, temperature and precipitation are the main factors affecting the distribution patterns of the microbial community [59–61]. However, in our study area, the elevational patterns of microbial community composition are likely to have been caused by soil P fractions. We will focus on analyzing the relationship between the microbial community (based on microbial PLFAs) and P fractions in the following text.

**Table 3. Spearman rank correlation coefficients between the soil P fractions (mg/kg) and the biomarker-PLFAs (%) measured from soil samples (n = 27).**

| Biomarker-PLFAs | Resin-Pi | NaHCO$_3$-Pi | NaHCO$_3$-Po | NaOH-Pi | NaOH-Po | $^{\Psi}$HCl-Pi | $^{\xi}$HCl-Po |
|---|---|---|---|---|---|---|---|
| **Bacteria** | 0.67$^{***}$ | 0.50$^{**}$ | 0.58$^{**}$ | -0.37 | -0.361 | -0.51$^{**}$ | -0.09 |
| **Fungi** | 0.45$^*$ | 0.70$^{***}$ | 0.67$^{***}$ | 0.02 | 0.10 | -0.24 | 0.08 |
| **Actinomycetes** | -0.35 | -0.24 | -0.36 | -0.15 | 0.49$^{**}$ | 0.60$^{**}$ | 0.05 |
| **AMF** | -0.44$^*$ | -0.34 | -0.40$^*$ | 0.15 | 0.34 | 0.57$^*$ | -0.01 |
| **Gram-positive bacteria** | 0.37 | 0.21 | 0.30 | 0.05 | -0.48$^*$ | -0.51$^{**}$ | -0.03 |
| **Gram-negative bacteria** | -0.13 | -0.08 | -0.139 | -0.286 | 0.42$^*$ | 0.45$^*$ | -0.03 |
| **cy:pre** | 0.43$^*$ | 0.49$^{**}$ | 0.52$^{**}$ | -0.19 | -0.39$^{**}$ | -0.68$^{***}$ | 0.07 |
| **Total PLFA** | -0.17 | -0.19 | -0.21 | -0.04 | 0.20 | 0.30 | 0.17 |
| **ACP** | -0.59$^{**}$ | -0.46$^*$ | -0.50$^{**}$ | -0.14 | 0.46$^*$ | 0.44$^*$ | 0.03 |

$^{\Psi}$HCl-Pi = HCl-Pi + conc.HCl-Pi,

$^{\xi}$HCl-Po = HCl-Po + conc.HCl-Po.

$^*$denotes significance at the 0.05 probability level;

$^{**}$denotes significance at the 0.01 probability level,

$^{***}$denotes significance at the 0.001 probability level.

## Microbial community and soil P fractions

The PLFA analysis provided comprehensive information on the responses of the microbial composition and the physiological state to the environmental gradients [48]. The RDA enabled us to explain the distribution patterns of microbial PLFAs using environmental factors. In this study, the measured environmental variables explained 73% of the variables in the microbial PLFAs. Additionally, the significant trend of the stress indicator (cy17:0/16:1 ω7) implied that the microbial communities were subjected to environmental stresses.

Generally, soil pH, as a single variable, has a more significant effect on the overall composition and diversity of the microbial community than temperature, latitude and geographic distance, especially at large spatial scales [3]. Previous studies have also shown that soil pH is the most important factor driving the microbial PLFA composition [13, 16]. However, our study showed that the effects of soil pH are not significant on the altitudinal pattern of microbial community distribution (Fig 2). The reason for the absence of any major role of pH could be the presence of a very narrow range of pH [60]. For this study, it was true that the soil pH was within a very narrow range and did not have a significant difference between elevations ($p > 0.05$, Table 1). To some extent, soils with similar pH values are beneficial for the elimination of the pH effect to evaluate the effects of other environmental factors.

In the present study, the RDA showed that gradients of P fractions played a much more important role than other factors (e.g., MAP, MAT, TOC, TN, pH and soil moisture). This finding meant that soil P fractions controlled the distribution pattern of soil microbial PLFAs with increasing elevation (Fig 2). Although previous studies across vegetation zones have shown that the distribution of microbial communities is mainly affected by soil carbon and nitrogen [62, 63], our results were not so. This mainly occurs for two reasons. First, each sampling point in our study area is within the coniferous forest zone of Shergyla Mountain; except for P fractions, other factors have little variability and are unlikely to dominate the distribution of the microbial community. Our results also showed that there was no significant difference in these factors between elevations (Table 1). The partial RDA indicated that the pure effect of these factors (17%) accounted for much smaller parts of the variability in PLFA composition than the pure effect of P fractions (40%) (Fig 2B). Second, P is likely to be the major limiting nutrient for microbial communities in this study area. In terms of soil TP concentration, TP in Shergyla Mountain (378−809 mg/kg, Table 1) is at a low level compared with Gongga Mountain (1180−1380 mg/kg) [57]. In particular, Augusto et al. [64] pointed out that when the P concentration in the solution is less than 45 μM (i.e., 1.4 mgP/kg), it indicates severe P deficiencies. In contrast, in our study, as a measure of P in soil solution, the Resin-Pi concentration ranged from 0.47 to 0.57 mg/kg. According to Augusto et al.'s standard, the soils are in a P-deficient state. Additionally, our results showed that bacterial, fungal and AMF PLFAs were significantly correlated with Resin-Pi (Table 3). There were significantly negative correlations between bioavailable P fractions and acid phosphatase, suggesting microbial mineralization of organic P was subjected to P stresses [65]. As an indicator of environmental stress, cy:pre (cy17:0/16:1 ω7) also showed a high correlation with bioavailable P fractions (i.e., Resin-Pi, $NaHCO_3$-Pi and $NaHCO_3$-Po, Table 3). Therefore, it is reasonable to believe that the distribution pattern of microorganisms is controlled by P fractions.

In addition, the $NaHCO_3$-Po present in the soil can, to some extent, be attributed to soil microorganisms. P released from dead microorganisms can be transformed into $NaHCO_3$-Po. For example, the release of soluble P, orthophosphate monoester and diester are much greater in autoclaved soils than in control soils [66]. Additionally, Louche *et al. [67]* reported that soil autoclaving could increase the organic P concentration extracted with sodium bicarbonate by >50%. This result was attributed to dead microorganisms. At our studied sites, the P released

from microorganisms obviously occurred because of a dynamic equilibrium between microbial death and reproduction in the microbial community. In a large community, it is likely that there is a large amount of P released by the large number of dead microorganisms. This released P can then be transformed into a considerable amount of $NaHCO_3$-Po. However, the P supply for microbial reproduction is likely derived from Resin-Pi and $NaHCO_3$-Pi. This is a likely explanation for why both bacterial and fungal PLFAs were positively correlated with $NaHCO_3$-Po ($r > 0.4$, $p < 0.05$). However, $NaHCO_3$-Po is labile and easily mineralized into inorganic P, which is quickly absorbed by soil organisms. In addition, a limited amount of dissolved organic P may be directly absorbed [68]. These processes can reduce the concentration of $NaHCO_3$-Po. It is possible that a large amount of $NaHCO_3$-Po may support a large amount of microorganisms. Furthermore, our results display a weakly negative correlation of $NaHCO_3$-Po with actinomycetes and AMF. Therefore, we suggest that the relationship between microorganisms and $NaHCO_3$-Po is complicated.

## Conclusions

This study demonstrated that microbial community composition varies with elevation and that the variation is markedly controlled by P fractions. Many biomarker PLFAs demonstrated that there were biogeographical trends of the microbial distribution patterns for soil microbial communities at different sites, which were often increasing, U-shaped and unimodal trends along the elevation gradient. Microorganisms are strongly associated with P fractions. Our results suggested that the microbial community was subjected to P stresses and that bioavailable P shaped the distribution pattern of microbial communities along the elevation gradient. Our work also implied that P geochemical processes drive microbial diversity. Further work is needed, including more experiments and field investigations, to determine the link between microbial community structure and P in alpine and subalpine terrestrial ecosystems.

## Supporting information

**S1 Table. The relative abundance of phospholipid fatty acids (PLFAs, %) across alpine and subalpine zones of Shergyla Mountain.**
(XLSX)

## Author Contributions

**Conceptualization:** Duo Ba, Yang Wang.

**Data curation:** Duo Ba, Duoji Qimei, Wei Zhao, Yang Wang.

**Formal analysis:** Duo Ba, Duoji Qimei.

**Funding acquisition:** Duoji Qimei, Yang Wang.

**Investigation:** Duo Ba, Duoji Qimei, Wei Zhao, Yang Wang.

**Methodology:** Duo Ba, Duoji Qimei, Wei Zhao.

**Project administration:** Duoji Qimei, Yang Wang.

**Supervision:** Duoji Qimei, Yang Wang.

**Validation:** Wei Zhao.

**Writing – original draft:** Duo Ba.

**Writing – review & editing:** Duoji Qimei, Wei Zhao, Yang Wang.

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
