## [Decision Letter · Decision Letter 0]

18 Apr 2022

PONE-D-22-06081Patterns of microbial communities were shaped by bioavailable P along the elevation gradient of Shergyla Mountain, by analysis of phospholipid fatty acidPLOS ONE

Dear Dr. wang,

Thank you for submitting your manuscript to PLOS ONE. After careful consideration, we feel that it has merit but does not fully meet PLOS ONE’s publication criteria as it currently stands. Therefore, we invite you to submit a revised version of the manuscript that addresses the points raised in the process.

We look forward to receiving your revised manuscript.

Kind regards,

Erika Kothe

Academic Editor

PLOS ONE

Journal Requirements:

3. In your Methods section, please provide additional information regarding the permits you obtained to collect samples for the present study. Please ensure you have included the full name of the authority that approved the field site access and, if no permits were required, a brief statement explaining why.

4. Thank you for stating the following in the Funding Section of your manuscript: 

"This work was funded by intergovernmental cooperation in science, technology and innovation, and by the National Key Research and Development Program (2019YFE0194000), the National Natural Science Foundation of China (42077013)."

We note that you have provided funding information. However, funding information should not appear in the Funding section or other areas of your manuscript. We will only publish funding information present in the Funding Statement section of the online submission form. 

"This work was funded by intergovernmental cooperation in science, technology and innovation, and by the National Key Research and Development Program (2019YFE0194000), the National Natural Science Foundation of China (42077013)."

Additional Editor Comments:

Both reviewers specifically commented on the shortcomings in language. Please revise accordingly with the help of a native speaker.

Reviewers' comments:

Reviewer's Responses to Questions

**Comments to the Author**

1. Is the manuscript technically sound, and do the data support the conclusions?

Reviewer #1: Yes

Reviewer #2: Yes

2. Has the statistical analysis been performed appropriately and rigorously? 

Reviewer #1: Yes

Reviewer #2: Yes

3. Have the authors made all data underlying the findings in their manuscript fully available?

Reviewer #1: Yes

Reviewer #2: Yes

4. Is the manuscript presented in an intelligible fashion and written in standard English?

Reviewer #1: Yes

Reviewer #2: Yes

5. Review Comments to the Author

Reviewer #1: The manuscript presented a case study of microbial community variation along elevation gradient in south-eastern Tibetan Plateau. This is a nice work of soil microbial diversity, especially from the special alpine region.

There are some language mistakes including authors' affiliations.

Reviewer #2: The study under review dealt with deciphering the relationship of P fractions and elevation gradients of mountain habitat of subalpine and alpine sites of Tibet, China on PLFA microbial community composition. The study determined the distribution pattern of PLFA biomarker microbial communities in relation to P availability prevailing across the gradient of mountain elevation. The understanding of this study may help in predicting the PLFA microbial communities and P availability while undertaking the management and rejuvenating the undisturbed soils/ecosystems. Overall study is nicely planned and executed and could be a good contribution in the science of microbial ecology. While going through the draft MS I noticed many shortcomings in relation of English language and typo errors which should be corrected while revising the MS legitimately.

Please check the detailed comments submitted to editor

6. PLOS authors have the option to publish the peer review history of their article (what does this mean?). If published, this will include your full peer review and any attached files.

Reviewer #1: **Yes: **Yanhong Wu

Reviewer #2: **Yes: **Dr Mahaveer P Sharma

---

## [Author Response · Author response to Decision Letter 0]

25 May 2022

Dear Editor and Referee:

We have studied the valuable comments from you and referees carefully, and tried our best to revise the manuscript (Manuscript Number: PONE-D-22-06081). The point to point replies to the referees’ comments are listed as following:

Reply to Journal Requirements:

Comment 1:

Reply:

According to the PLOS ONE style templates, our manuscript has been revised. We ensure that our manuscript meets PLOS ONE's style requirements. 

Comment 2:

Reply:

We have asked a language edit company (American Journal Experts, see the Editing Certificate in our covering letter) to help us to improve our manuscript. In its present form, we believe that the English quality is at the standard that is required for publication. In particular, in order to ensure correct grammar, the title is changed as follows: Patterns of microbial communities were shaped by bioavailable P along the elevation gradient of Shergyla Mountain, as determined by analysis of phospholipid fatty acids.

Comment 3:

3. In your Methods section, please provide additional information regarding the permits you obtained to collect samples for the present study. Please ensure you have included the full name of the authority that approved the field site access and, if no permits were required, a brief statement explaining why.

Reply:

According to the comment, in your Methods section, we provided additional information regarding the permits we obtained to collect samples for the present study. See line 109-115 in the file " Revised Manuscript with Track Changes ".

Ethics Statement

All necessary permits were obtained for the described field studies. We conducted this study at the Shergyla Mountain, which is under the jurisdiction of the people's Government of Nyingchi City, Tibet Autonomous Region, people's Republic of China. We obtained permission from the people's Government of Nyingchi City to use the sample plots. Furthermore, our study did not harm the environment and did not involve endangered or protected species.

Comment 4:

4. Thank you for stating the following in the Funding Section of your manuscript: 

"This work was funded by intergovernmental cooperation in science, technology and innovation, and by the National Key Research and Development Program (2019YFE0194000), the National Natural Science Foundation of China (42077013)."

We note that you have provided funding information. However, funding information should not appear in the Funding section or other areas of your manuscript. We will only publish funding information present in the Funding Statement section of the online submission form. 

"This work was funded by intergovernmental cooperation in science, technology and innovation, and by the National Key Research and Development Program (2019YFE0194000), the National Natural Science Foundation of China (42077013)."

Reply:

We removed any funding-related text from the manuscript. Our Funding Statement is as follows: This work was funded by intergovernmental cooperation in science, technology and innovation, and by the National Key Research and Development Program (2019YFE0194000), the National Natural Science Foundation of China (42077013).

Moreover, the Funding Statement is included within our cover letter.

Comment 5:

5. Please include your full ethics statement in the ‘Methods’ section of your manuscript file. In your statement, please include the full name of the IRB or ethics committee who approved or waived your study, as well as whether or not you obtained informed written or verbal consent. If consent was waived for your study, please include this information in your statement as well..

Reply:

According to the comments, we provided the full ethics statement in the ‘Methods’ section of our manuscript file. See line 109-115 in the file "Revised Manuscript with Track Changes".

 

Reply to the Additional Editor Comments

Comment 1:

Both reviewers specifically commented on the shortcomings in language. Please revise accordingly with the help of a native speaker.

Reply:

We have asked a language edit company (American Journal Experts, see the Editing Certificate in our covering letter) to help us to improve our manuscript. In its present form, we believe that the English quality is at the standard that is required for publication. The Editing Certificate is in the file "Response to Reviewers".

Reply to the Reviewers' comments:

Comment 1:

.1. Is the manuscript technically sound, and do the data support the conclusions?

Reviewer #1: Yes

Reviewer #2: Yes

Reply:

Thank the reviewers for their approval.

Comment 2:

2. Has the statistical analysis been performed appropriately and rigorously?

Reviewer #1: Yes

Reviewer #2: Yes.

Reply:

Thank the reviewers for their approval.

Comment 3:

3. Have the authors made all data underlying the findings in their manuscript fully available?

Reviewer #1: Yes

Reviewer #2: Yes.

Reply:

Thank the reviewers for their approval.

Comment 4:

4. Is the manuscript presented in an intelligible fashion and written in standard English?

Reviewer #1: Yes

Reviewer #2: Yes

Reply:

Thank the reviewers for their approval.

Comment 5:

5. Review Comments to the Author

Reviewer #1: The manuscript presented a case study of microbial community variation along elevation gradient in south-eastern Tibetan Plateau. This is a nice work of soil microbial diversity, especially from the special alpine region.

There are some language mistakes including authors' affiliations.

Reviewer #2: The study under review dealt with deciphering the relationship of P fractions and elevation gradients of mountain habitat of subalpine and alpine sites of Tibet, China on PLFA microbial community composition. The study determined the distribution pattern of PLFA biomarker microbial communities in relation to P availability prevailing across the gradient of mountain elevation. The understanding of this study may help in predicting the PLFA microbial communities and P availability while undertaking the management and rejuvenating the undisturbed soils/ecosystems. Overall study is nicely planned and executed and could be a good contribution in the science of microbial ecology. While going through the draft MS I noticed many shortcomings in relation of English language and typo errors which should be corrected while revising the MS legitimately.

Please check the detailed comments submitted to editor.

Reply:

To improve the English language of our manuscript, we have asked a language edit company (American Journal Experts) to help us. In its present form, we believe that the English quality is at the standard that is required for publication.

---

## [Editor Report · Decision Letter 1]

24 Jun 2022

Patterns of microbial communities were shaped by bioavailable P along the elevation gradient of Shergyla Mountain, as determined by analysis of phospholipid fatty acids

PONE-D-22-06081R1

Dear Dr. wang,

We’re pleased to inform you that your manuscript has been judged scientifically suitable for publication and will be formally accepted for publication once it meets all outstanding technical requirements.

Kind regards,

Erika Kothe

Academic Editor

PLOS ONE
---

## [Editor Report · Acceptance letter]

30 Jun 2022

PONE-D-22-06081R1 

Patterns of microbial communities were shaped by bioavailable P along the elevation gradient of Shergyla Mountain, as determined by analysis of phospholipid fatty acids 

Dear Dr. Wang:

I'm pleased to inform you that your manuscript has been deemed suitable for publication in PLOS ONE. Congratulations! Your manuscript is now with our production department. 

Kind regards, 

on behalf of

Prof. Dr. Erika Kothe 

Academic Editor

PLOS ONE